# Mechanical properties of discarded shield residue improved by calcium carbide slag and fly ash as subgrade filling

**Shoujie Wang, Jianwen Ding** *, **Qingying Guo, Ning Jiao, Chenhao Li**

Institute of Geotechnical Engineering, School of Transportation, Southeast University, Nanjing, China

* jwding@seu.edu.cn

## Abstract

To utilize discarded shield residue and alleviate the shortage of subgrade filling, industrial wastes such as calcium carbide slag (CCS) and fly ash (FA) were considered to enhance the mechanical properties of the shield residue. A series of laboratory tests, including California Bearing Ratio (CBR) tests, unconfined compressive strength (UCS) tests, moisture content tests, pH tests, water stability tests, and dry-wet cycles tests were performed on discarded shield residue with additive contents. The results show that the UCS and CBR values enhanced significantly with the increase in curing time. However, the moisture content and pH of the stabilized soil exhibited a decreasing trend. The early UCS of CCS-FA stabilized soil is slightly lower than that of QL-FA stabilized soil. After 60 curing days, all stabilized soil exhibited a UCS value exceeding 1.9 MPa. In addition, the CBR values of CCS-FA stabilized soil were more than 8 times higher than those of the original shield residue. Furthermore, the water stability of CCS-FA stabilized soil is slightly better than QL-FA stabilized soil, especially at 7 days and 14 days. As for dry-wet cycles test, after the fifth cycle, the CCS-FA stabilized soil maintained overall integrity. The CCS can effectively replace QL to enhance the mechanical properties of shield residue as subgrade filling.

## 1. Introduction

The shield tunneling method is commonly used in metro construction [1], producing substantial shield residue [2]. The high moisture content, high porosity, and high disposal cost of the shield residue make it become the predominant source of municipal solid waste [3,4]. It is necessary to explore appropriate methods to improve the engineering adaptability of discarded shield residue. Quicklime (QL) is commonly adopted to reduce the moisture content, expansion deformation, and to enhance the strength of residue [5–8]. However, due to the high cost and ineffectiveness in water stability [9], several industrial by-products (IBPs), such as fly ash (FA), rice husk ash (RHA), and ground-granulated blast-furnace slag (GGBS), are widely used as supplementary cement materials (SCMs) for soil improvement [10–12]. The use of SCMs with lime-based compounds can not only save resources but also enhance the mechanical properties and durable of the stabilized soil [13,14].

**Funding:** This study is partially supported by the National Natural Science Foundation of China (Grant No. 52378330 and Grant No. 51978159). The funders had no role in study design, data collection and analysis, decision to publish, or preparation of the manuscript.

**Competing interests:** All authors have declared that they have no financial interests that could be perceived as influencing the work presented in this paper

CCS is generated from the hydrolysis of calcium carbide during acetylene production. The main composition is similar to hydrated lime [15,16]. Hence, the improvement mechanism of CCS is akin to QL. When mixed with pozzolanic materials, it can form cementitious material such as calcium silicate hydrate [17,18]. The CCS has been widely utilized in clay, expansive soil, and loess soil improvement. The feasibility of using CCS and FA to improve the silty clay was validated, and the strength control mechanism for the stabilization of silty clay with CCS and FA was proposed (Horpibulsuk et al. [19,20]). Research indicated that unconfined compressive strength (UCS) of expansive soil increased by more than 10 times with the addition of CCS and FA (Noolu et al. [21]). The combined effects of CCS and FA as SCMs was analyzed and the effectiveness as a stabilizer for loess soil of CCS-FA binders was demonstrated (Julphunthong et al. [22]). Additionally, the strength of the stabilized soil with different CCS and FA contents was investigated, and it was found that the CCS: FA ratio of 40: 60 with 12% binder increased the strength of rammed earth significantly (Siddiqua et al. [23]). Furthermore, CCS was a substitute for QL as a subgrade filling (Du et al [24]). Proper utilization of CCS for subgrade filling soil stabilization is an important way of resource conservation (Jiao et al. [25]). In conclusion, the existing research mainly focuses on strength evaluation, while the water stability and durability of the stabilized soil are rarely discussed. Moreover, fewer studies have investigated the use of CCS and FA to improve shield residue. Further research is needed to investigate the mechanical properties of CCS-FA stabilized soil. It is significant in providing theoretical guidance for replacing QL with CCS in the enhancement of shield residue as subgrade filling.

This paper is based on the west extension of Jihua Road and the construction of shield tunnels in Foshan City, Guangzhou Province, China. The aim of this paper is to enhance the shield residue with IBPs (i.e. CCS and FA) and utilize it for subgrade filling during the construction of Jihua Road, in order to achieve resource utilization and cost savings. Firstly, a series of laboratory experiments, such as the California Bearing ratio (CBR) test, UCS test, moisture content test, pH test, water stability test, and dry-wet cycles test, were conducted. Then, the mechanical properties and engineering characteristics of CCS-FA stabilized soil were compared with QL-FA stabilized soil. Finally, the feasibility of replacing QL with CCS for enhancing shield residue as subgrade filling was analyzed. The investigation will provide a theoretical foundation and technical support for the improvement of shield residues and the application of IBPs.

## 2 Materials and methods

### 2.1 Testing materials

The shield residue was obtained from the tunnel excavation in Zhongshan Park section in Foshan. The basic physical properties of the shield residue are shown in Table 1. QL, CCS, and FA were taken from the construction site. The main oxide compositions of the shield residue and additive materials, determined by X-ray fluorescence spectroscopy (XRF), are listed in Table 2. The X-ray diffraction spectrum is shown in Fig 1. The XRF results indicated that the primary composition of the shield residue was $SiO_2$, with minor amounts of $CaCO_3$, $Al_2O_3$, and $Fe_2O_3$. The main composition of both CCS and QL was CaO. The primary composition of FA was SiO2, Al2O3, and Fe2O3, constituting approximately 78%. Based on the XRD patterns shown in Fig 1, the primary crystalline material in the shield residue was calcite ($CaCO_3$) and quartz ($SiO_2$). And the predominant crystalline material in both QL and CCS was portlandite ($Ca(OH)_2$). As for FA, the broad hump suggested that FA is mainly comprised of amorphous phases with some peaks of Chiastolite ($Al_2(SiO_2)O$), quartz, ferric oxide, and lime.

**Table 1. Basic physical parameters of shield residue.**

| Physical index | | Value |
|---|---|---|
| Initial Moisture content/% | | 59.1 |
| Density/(g/cm$^3$) | | 1.79 |
| Dry density/(g/cm$^3$) | | 1.12 |
| Particle distribution/% | <5μm | 15.1 |
| | 5–75μm | 27.8 |
| | >75μm | 57.1 |
| Liquid limit/% | | 44.8 |
| Plastic limit/% | | 21.4 |
| Plasticity index | | 23.4 |

## 2.2. Testing methods

The experimental mix proportions were designed based on the high liquid limit subgrade soil improvement method [26–28], The 3% QL (CCS) content was used as the benchmark, and the 3%, 4%, and 6% FA were added to verify the feasibility of substituting CCS for QL. The experimental mix proportions are shown in Table 3.

To determine the optimum water content and maximum dry density of the stabilized soil, the compaction test was conducted according to ASTM D1557 [29]. The specimens with 172 mm in height and 150 mm in diameter were compacted in three layers, and each layer was given 98 hits by a 4.5 kg hammer dropped from a height of 45 cm.

The specimens were prepared at the optimal moisture content with 96% compaction. The CBR value and swelling rate were determined after 4 days immersed in water according to ASTM D1883-16a [30].

The UCS test was conducted using cylindrical specimens with dimensions of Φ50mm×50mm referencing ASTM D2166 [31]. The specimens were statically compacted from both sides of a compaction mold to get a compaction level of 96%. The water content was equal to the optimum water content, and the curing time was set at 7, 14, 28, and 60 days, with a temperature of 20±2˚C and humidity≥95%. To investigate the water stability of the stabilized soil, the ratio ($k$) of the UCS of the submerged specimens (saturated UCS) to non-submerged specimens (unsaturated UCS) at the same curing time was used as a reference indicator for the ability to maintain its original state under submerged conditions.

The pH tests were carried out in accordance with ASTM D4972 [32], whereas a solution with a soil-water ratio of 1:1 was prepared. After standing for 1 hour, the supernatant of the solution was used for pH measurement.

**Table 2. Primary composition of experimental materials.**

| element | Material | | | |
|---|---|---|---|---|
| | Shield residue | QL | CCS | FA |
| SiO$_2$ | 49.26 | 3.26 | 4.98 | 40.15 |
| CaO | 15.81 | 93.15 | 92.34 | 12.78 |
| Al$_2$O$_3$ | 13.07 | 0.46 | 1.94 | 22.74 |
| Fe$_2$O$_3$ | 12.02 | - | 0.14 | 14.70 |
| K$_2$O | 7.20 | - | 0.25 | 1.25 |
| SO$_3$ | - | 0.81 | 0.15 | 3.08 |
| TiO$_2$ | 1.34 | - | - | 3.49 |

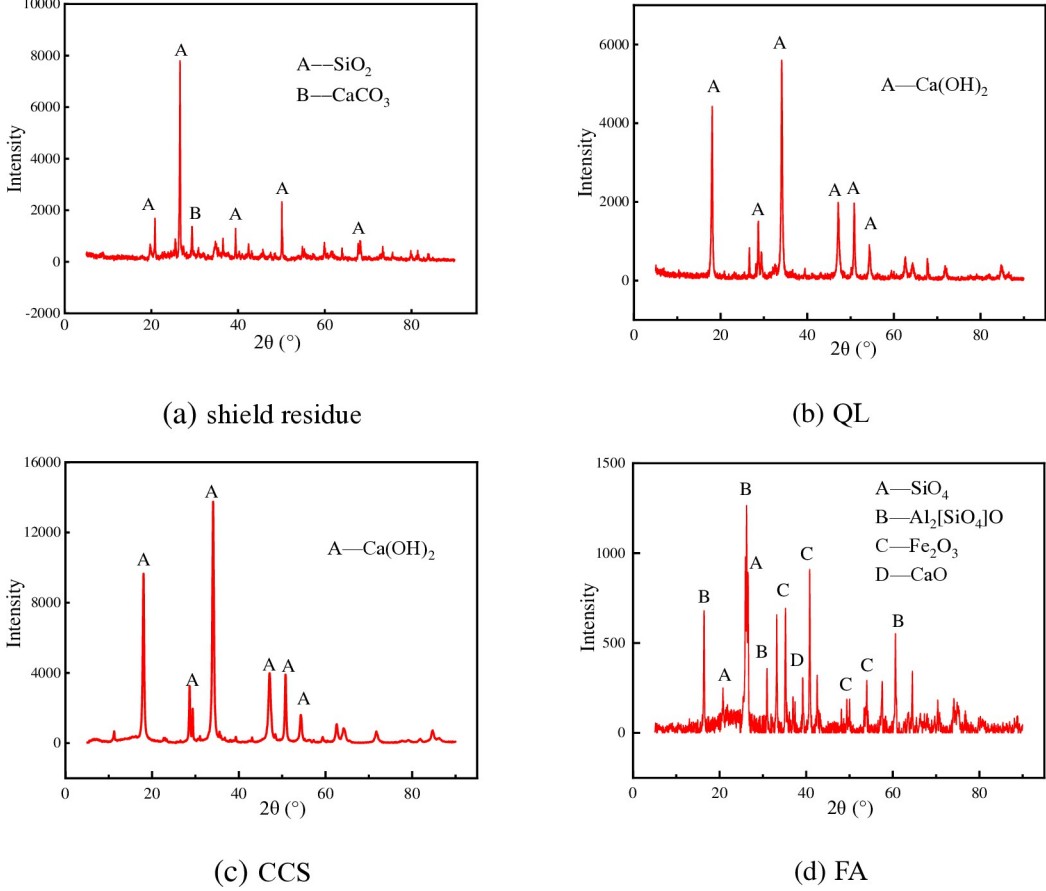

**Fig 1. XRD diffraction spectrum of experimental material.**

The dry-wet cycle tests were conducted on specimens cured for 28 days under standard conditions, with sample preparation and strength testing methods referenced from the UCS tests and similar research [6,33]. Each cycle involved drying for 16 hours and then wetting in water for 8 hours. This process was repeated 5 times to complete 5 dry-wet cycles.

## 3. Results and discussion

### 3.1 Compaction test

The compaction test results are shown in Table 4. It could be observed that after adding improved materials to the shield residue, the optimum moisture content of the stabilized soil

**Table 3. Experimental mix proportions.**

| Specimens | | QL (%) | CCS (%) | FA (%) |
|---|---|---|---|---|
| A | QLFA3 | 3 | | 3 |
| | QLFA4 | 3 | | 4 |
| | QLFA6 | 3 | | 6 |
| B | CCSFA3 | | 3 | 3 |
| | CCSFA4 | | 3 | 4 |
| | CCSFA6 | | 3 | 6 |

**Table 4. Compaction test results.**

| Specimens | | Optimum moisture content (%) | Maximum dry density(g·cm$^{-3}$) |
|---|---|---|---|
| Shield residue | | 16.9 | 1.72 |
| A | QLFA3 | 21.3 | 1.58 |
| | QLFA4 | 22.0 | 1.57 |
| | QLFA6 | 22.1 | 1.57 |
| B | CCSFA3 | 21.8 | 1.59 |
| | CCSFA4 | 22.1 | 1.56 |
| | CCSFA6 | 22.2 | 1.55 |

increased, while the maximum dry density decreased somewhat. This might be attributed to the lower density of the improved material compared to shield residues, as well as some flocculent cementation products that result in a certain volume expansion, thereby reducing soil compactness and density [26,34]. Conversely, the incorporation of FA could fill soil pores and the greater specific surface area of FA could facilitate the formation of bound water as water molecules infiltrate the micropores [15]. As a result, the optimal water content of stabilized soil increased with the increase of FA [20,35]. Besides, FA was prone to pozzolanic reactions with QL and CCS to generate cementitious materials. This reaction consumed a large amount of water, implying additional water needs to be added in order to obtain the stabilized soil with optimal water content. Consequently, as the FA content increased, the optimum moisture content of the stabilized soil also increased [16]. Notably, the optimum moisture content and maximum dry density of the CCS-FA stabilized soil were essentially the same as QL-FA stabilized soil with the same FA content. The phenomenon indicated that the compaction characteristics of the stabilized soil remained unchanged when using CCS as a substitute for QL.

## 3.2 California bearing ratio test

The CBR value is the main index to evaluate the bearing capacity of the subgrade. Fig 2 shows the swelling volume and CBR values of the stabilized soil. The CBR value and swelling rate of the raw shield residue were 8.1% and 3.0%, respectively. The corresponding CBR values of QLFA3, QLFA4, and QLFA6 were 109.3%, 119.3%, and 125.6%, which were 13.5 times, 14.7 times, and 15.5 times of the original shield residue, respectively. Meanwhile, due to the hydration reaction between the improved materials mixed in the shield residues, providing cementitious substances to fill the pores between soil particles and enhance the cohesion of the soil, the compressive strength and bearing capacity of the stabilized soil increases accordingly. As a result, the CBR value of QL-FA stabilized soil increased with the FA content. Similarly, the CBR values of the CCSFA3, CCSFA4, and CCSFA6 were 66.6%, 72.8%, and 86.3%, which were 8.2 times, 9.0 times, and 10.7 times of the original shield residue, respectively. These CBR values were slightly lower than those of the QL-FA stabilized soil at the same FA content. It also could be observed that as the FA content increased, there was a corresponding increase in the CBR values of the CCS-FA stabilized soil. This trend suggested that the incorporation of FA could also enhance the bearing capacity of CCS-FA stabilized soil. Furthermore, the swelling rate of all stabilized soil decreased by more than 70% and remained below 1.0%. The swelling rate of the QL-FA stabilized was slightly lower compared to CCS-FA stabilized soil, but the difference between the two is small. This indicated that the swelling trend of both stabilized soils was not significantly different.

To investigate the impact of curing time on the CBR values, specimens from the QLFA4 and CCSFA4 were cured for 7 days and 28 days, respectively, and then submerged for four

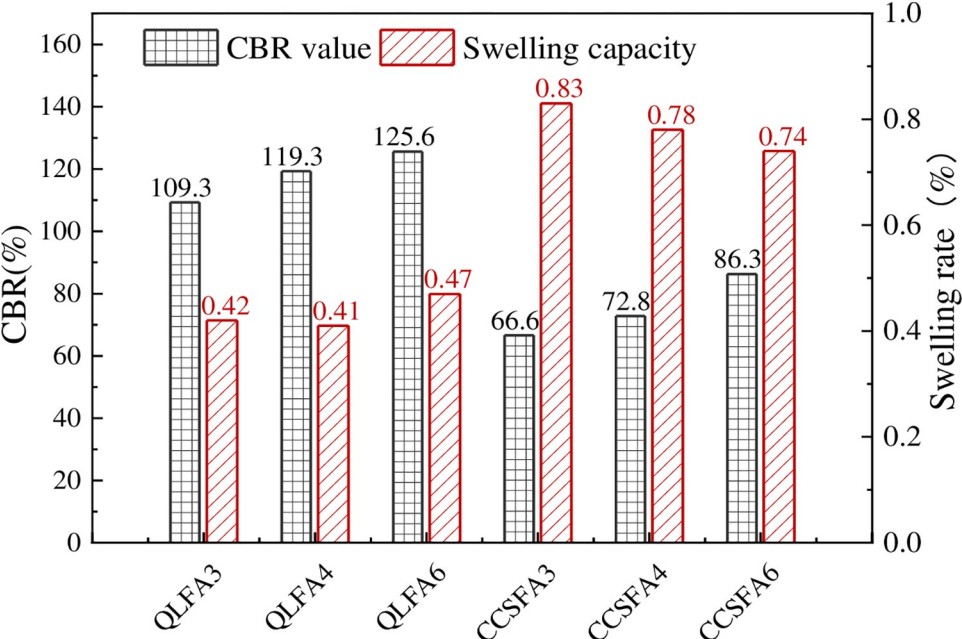

**Fig 2. CBR test results.**

days before CBR tests. As shown in Fig 3, there was a remarkable increase in CBR values for both stabilized soils. The specimens of CCSFA4 exhibited an increase rate of 49.2% at 7 days and further expanded to 190.25% at 28 days. Similarly, the increase rate of specimens from

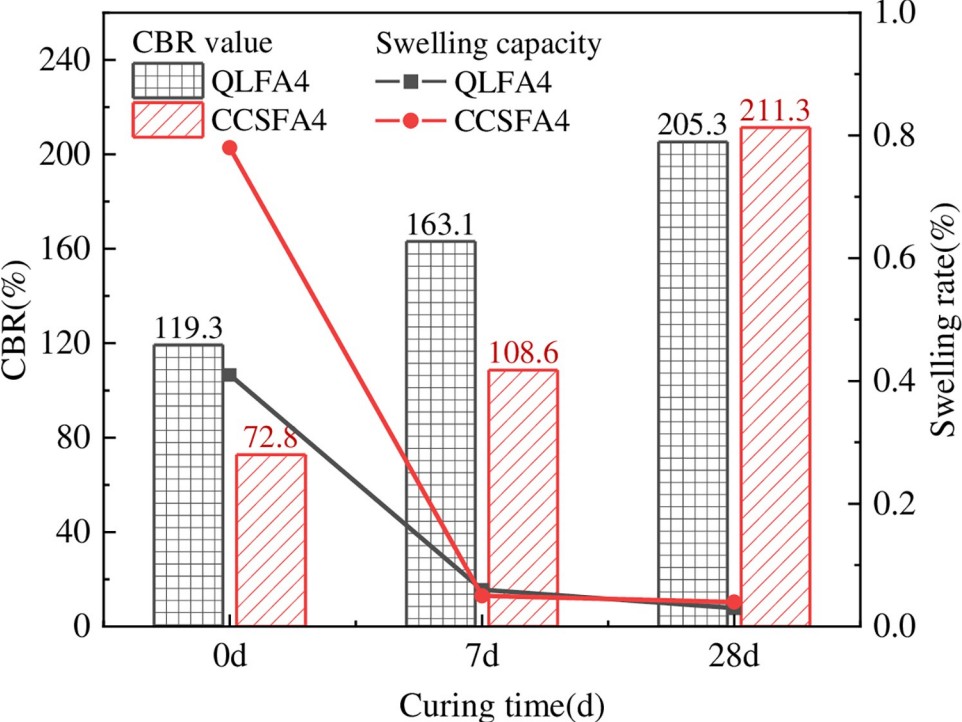

**Fig 3. CBR test results at different curing time.**

QLFA4 was 36.66% and 72.1% at 7 and 28 days, respectively. Initially, the CBR values of the CCS-FA stabilized soil were slightly lower than that of the QL-FA stabilized soil. However, due to its higher strength growth rate, especially after 7 days of curing, the CBR values exceeded QL-FA stabilized soil slightly at 28 days. In addition, the swelling rate of both stabilized soils substantially decreased, reducing to within 0.1% after 7 days of standard curing.

### 3.3 Unconfined compressive strength test

The relationship between the UCS and curing time is shown in Fig 4. The UCS of the raw shield material specimen was 0.44 MPa. After the incorporation of improved materials, the UCS of the shield residue was significantly enhanced. At the same mix proportions, the UCS of the stabilized soil increased with curing time. When curing time from 7 days to 28 days, the UCS of QL-FA stabilized soil increased by 1.30 times approximately, and that of CCS-FA stabilized soil increased by 1.23 times approximately. After 28 days of curing, the strength growth slowed down, and the reaction tended to stabilize. This was primarily due to the flocculation and hydration reaction in the initial period. The CaO in the additive could hydrate to form $Ca(OH)_2$, which improves the UCS after carbonization and precipitation. Furthermore, $Ca(OH)_2$ reacted with the active $SiO_2$ and $Al_2O_3$ in FA to generate a large amount of calcium silicate hydrate (C-S-H) and calcium aluminate hydrate (C-A-H). These cementitious materials could form a dense spatial structure and enhance the strength of the stabilized soil. The reaction rate decreased gradually over the curing time, resulting in a deceleration of the strength development in the soil until it reached a steady state [20,21].

For QL-FA stabilized soil, the UCS was not significantly affected by FA content, as the curves for incorporating 4% and 6% FA almost overlapped. This phenomenon could be attributed to the formation of cementitious substances that aggregated the soil particles during the modification process. At low FA content, the active $SiO_2$ and $Al_2O_3$ present in FA helped supply the necessary silicon and aluminum components. Increasing the content of FA appropriately could enhance pozzolanic reaction, generating more cementitious substances. However,

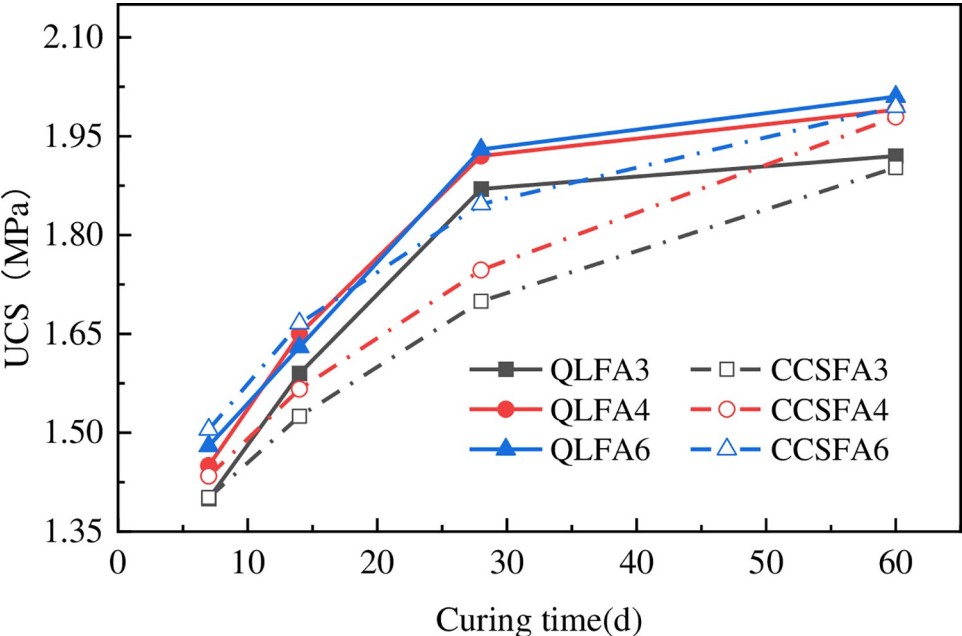

**Fig 4. Relationship between UCS and curing time.**

the UCS increased slowly due to the lack of sufficient hydroxide ions and calcium ions in the stabilized soil when FA content beyond a certain threshold.

The UCS for the stabilized soil exhibited closely similar trends. In the first 7 days of curing, the UCS of the specimens with the same FA content were essentially the same. Subsequently, the UCS growth of QL-FA stabilized soil demonstrated a relatively rapid rate from 7 to 28 days, and slightly surpassed that of the CCS-FA stabilized soil at 28 days. After 28 days, the strength growth of the QL-FA stabilized soil slowed down, while that of the CCS-FA stabilized soil continued to progress at a fast pace. The QL-FA stabilized soil exhibited an average strength increase of 16.0 kPa per day during the first 28 days, followed by a rate of 1.9 kPa from days 28 to 60. In contrast, the CCS-FA stabilized soil showed a growth rate of 11.4 kPa per day in the initial 28 days, which decreased but remained higher at 6.0 kPa per day from days 28 to 60. Therefore, until 60 days, the UCS of both stabilized soils became similar again, by margins of 0.02 MPa, 0.01 MPa, and 0.02 MPa, respectively. This phenomenon might be attributed to the more intense pozzolanic and ion exchange reactions occurring in the QL-FA stabilized soil during the initial stage. However, the higher pH value provided the necessary alkaline excitation environment, and the active components of CCS ($SiO_2$ at 4.88% and $Al_2O_3$ at 1.94%) were higher than those of QL ($SiO_2$ at 3.26% and $Al_2O_3$ at 0.46%), which led to a more repid subsequent strength growth of CCS-FA stabilized soil. When CCS was used to replace QL for soil improvement, the impact on the mechanical properties of the stabilized soil was relatively slight. There were only trivial differences in the strength growth process during the early curing period. Using CCS to replace QL for soil improvement exhibited high feasibility.

Fig 5 shows the schematic diagram of the damage process of the stabilized soil, which can be divided into three stages:

1. After a period of vertical loading, fine cracks appeared in the middle of the specimen's surface and gradually developed into V-shaped towards both ends.

2. With the continued vertical loading, the cracks increased and widened. The unconstrained middle part of the specimen exhibited greater lateral deformation and spalling.

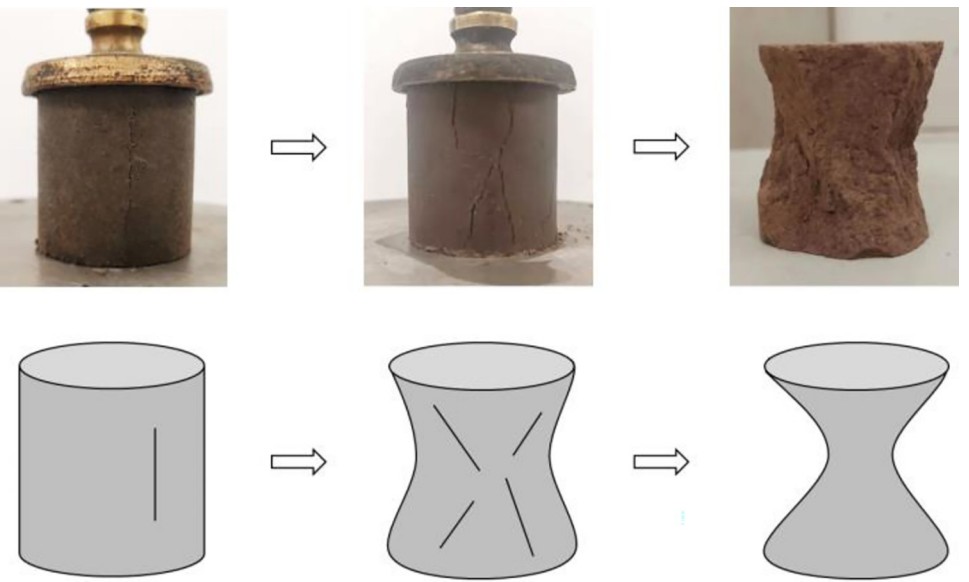

**Fig 5. Schematic diagram of the damage process of the stabilized soil.**

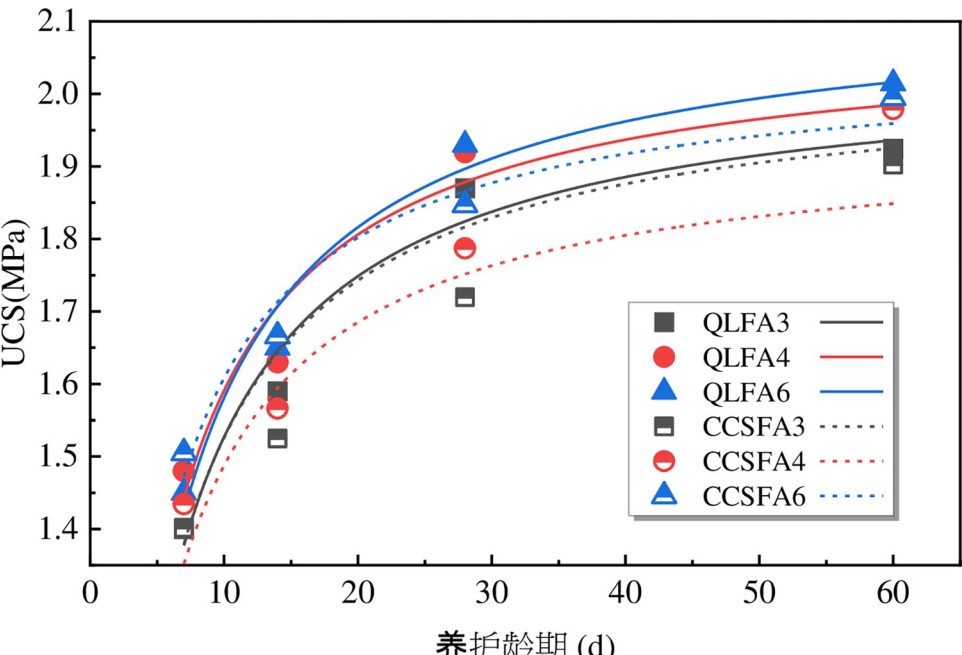

**Fig 6. Strength characteristic curve of stabilized soil.**

3. Transverse deformation of the specimen at the middle intensified, with spalling occurring around the entire specimen. The specimen presented a shape resembling two cones joined together. Consequently, it lost its bearing capacity and was destroyed.

From the initial appearance of fine cracks to the final failure, the whole process was short, indicating a brittle failure mode.

To delve deeper into the relationship between the curing time and the UCS, a fitting analysis was conducted on the UCS of each mix proportion. It could be noted that the UCS could converge to a hyperbolic function as shown in Eq (1).

$$q_u = \frac{p_1 t}{p_2 + t} \tag{1}$$

Where qu denotes the UCS of the specimen (MPa); $p_1$, $p_2$ denotes the regression parameters; t denotes the curing time (d).

The UCS characteristic curves with curing time are illustrated in Fig 6. The corresponding regression parameters are listed in Table 5 accordingly. It could be observed that there was a

**Table 5. Hyperbolic intensity characteristic curve regression parameters.**

| Specimens | | $p_1$ | $p_2$ | $R^2$ |
|---|---|---|---|---|
| A | QLFA3 | 2.04553 | 3.39633 | 0.95122 |
| | QLFA4 | 2.08822 | 3.1251 | 0.92169 |
| | QLFA6 | 2.13346 | 3.49453 | 0.96209 |
| B | CCSFA3 | 1.9432 | 3.06376 | 0.88443 |
| | CCSFA4 | 2.03114 | 3.30912 | 0.89023 |
| | CCSFA6 | 2.04864 | 2.73955 | 0.94606 |

good correlation between the UCS and the characteristic curves with $R^2$ values all exceeding 0.88. The hyperbolic function could effectively fit the characteristic curves of UCS as it varies with the curing time. Analysis of the regression parameters revealed that the variation range for parameter p1 is relatively small and was not significantly affected by the type or content of the improved materials, remaining between 1.94 and 2.14. In contrast, variation in material content had a more significant impact on parameter p2, which ranges from 2.73 to 3.50.

### 3.4 Moisture content and pH test

The moisture content and pH value of the specimen reflect the progress of the hydration reaction to some extent. It could be seen from Figs 7 and 8 that both the moisture content and pH value of the stabilized soil tended to decrease with curing time. The trend of moisture content and pH change for all mix proportions was relatively consistent, exhibiting a significant decrease between 7 days and 28 days, followed by a gradual stabilization from 28 days to 60 days. The pH curves for the QL-FA stabilized soil were relatively close compared to the CCS-FA stabilized soil, showing that the change in FA content has less influence on it.

The main reasons for the decrease in pH value over the curing time were summarized as follows:

1. Upon contact with the soil's pore water, QL and CCS promptly reacted to generate a large amount of $Ca(OH)_2$.

2. A portion of the $Ca(OH)_2$ dissolved to ionize $Ca^{2+}$ and $OH^-$. The $OH^-$ then partially reacted with the $H^+$ to form $H_2O$, and the rest increased the pH of the pore water.

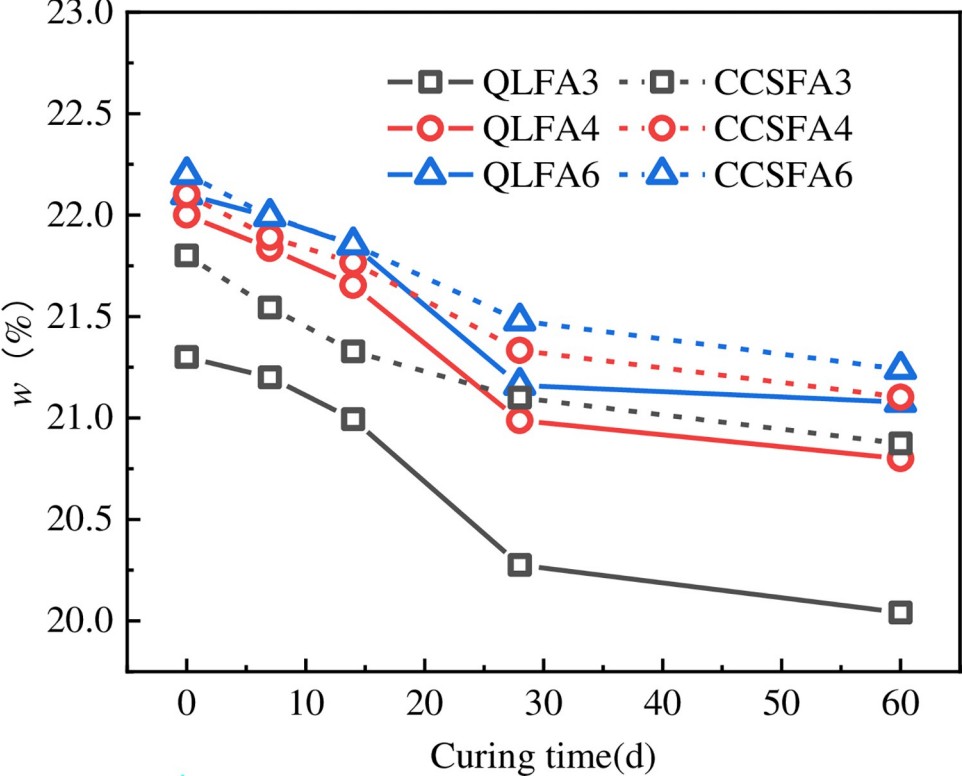

**Fig 7. Change in moisture content of stabilized soil.**

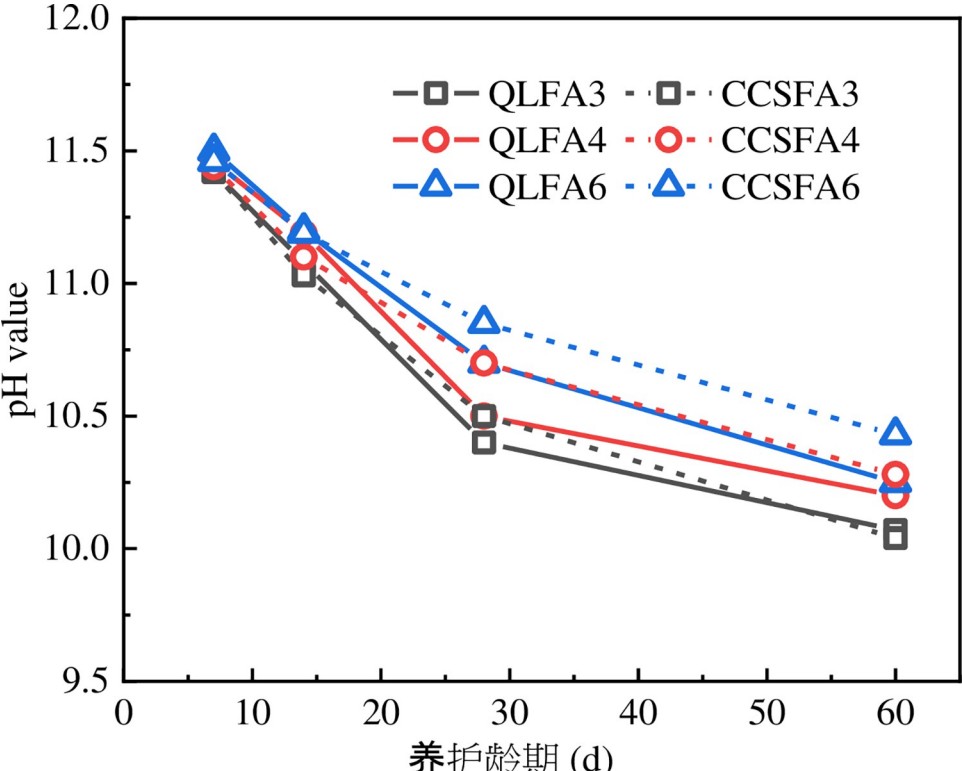

**Fig 8. Change in pH value of stabilized soil.**

3. As the pH value increased, the solubility of active $SiO_2$ and $Al_2O_3$ in the soil particles also rose, facilitating further reactions with $Ca^{2+}$ and $OH^-$. The continuous consumption of $OH^-$ ions in the pozzolanic reaction ultimately led to a decrease in the pH value of the stabilized soil.

## 3.5 Water stability test

Fig 9 illustrates the variation in UCS before and after immersion. It was evident that the UCS of all specimens generally decreased after immersion, and the most reduction was recorded at 7 days, exceeding 0.60 MPa. The comparison of saturated UCS between the two types of stabilized soils indicated the superior water stability of CCS-FA stabilized soil.

The relationship between the water stability coefficient ($k$) and curing time is shown in Fig 10. The figure intuitively reflects a gradual diminishment in strength degradation after soaking over time, indicating an improvement in the water stability of all specimens with the curing time. The $k$ significantly increased from 7 days to 28 days. The growth rates of QLFA3, QLFA4, and QLFA6 were 26.61%, 37.12%, and 35.79%, respectively, and those of CCSFA3, CCSFA4, and CCSFA6 were 24.63%, 21.84%, and 23.17%, respectively. The $k$ of all specimens was beyond 60% after 28 days of curing. Under the same curing time, the $k$ of the CCS-FA stabilized soil was slightly higher than that of QL-FA stabilized soil with the same FA content. Especially at 7 days and 14 days, the $k$ of the CCS-FA stabilized soil was approximately 1.05 to 1.26 times that of QL-FA stabilized soil. This phenomenon could own to the finer particle size and larger specific surface area of CCS, which could lead to a denser microstructure of the stabilized soil. The addition of CCS reduces water permeation and strength loss of stabilized soil

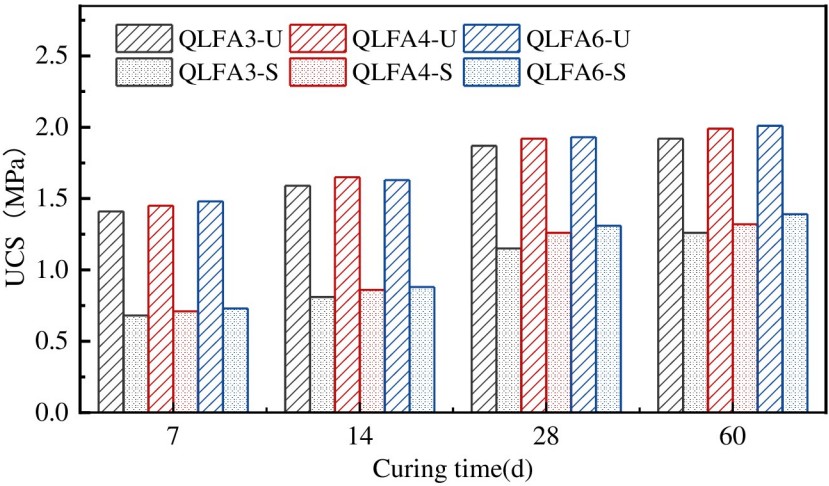

(a) QL-FA stabilized soil

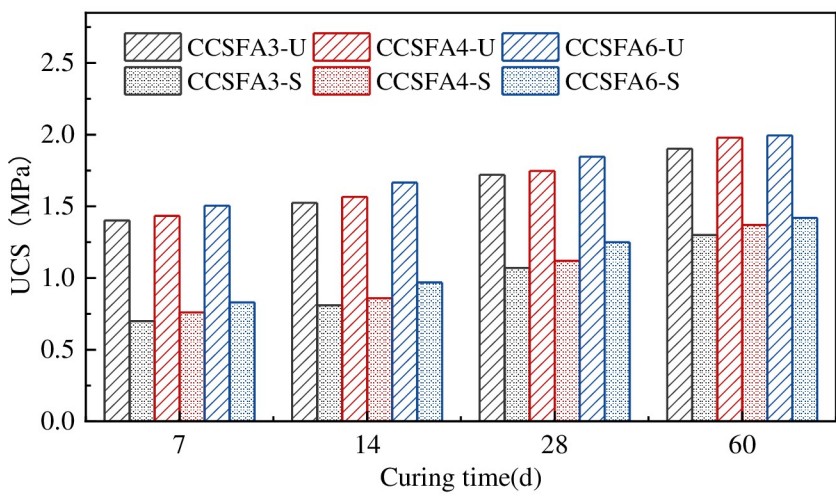

(b) CCS-FA stabilized soil

**Fig 9. Comparison of saturated and unsaturated UCS of stabilized soil.**

[19], further confirming the feasibility of utilizing CCS to replace QL in shield residue improvement.

## 3.6 Dry-wet cycle test

Foshan is located in a coastal area with high temperatures and heavy rainfall during the summer. Due to seasonal variations in precipitation and evaporation, the subgrade is subjected to cycles of being saturated and unsaturated over time. It is necessary to take into account the effects of multiple dry-wet cycles on the strength of subgrade materials. The UCS of QL-FA stabilized soil with 4% and 6% FA incorporation waswa almost the same. And the CBR value of QL-FA stabilized soil with 6% FA incorporation was only slightly higher than that of stabilized soil with 4% FA incorporation. In line with economic considerations, the lower total

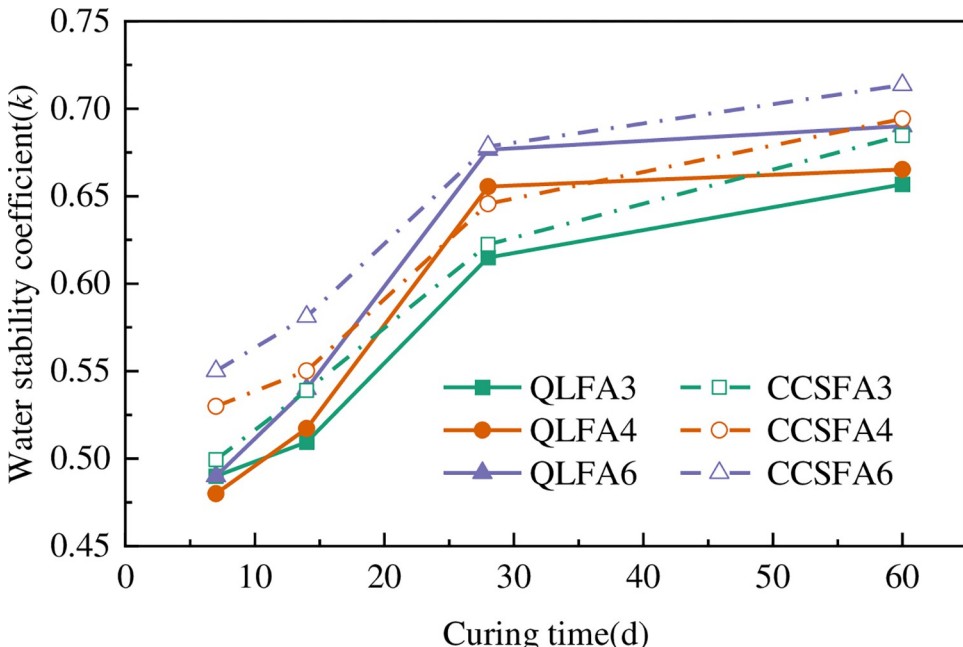

**Fig 10. Relationship between water stability coefficient and curing time.**

dosage of 3% QL and 4% FA (QLFA4) was selected for further testing. The corresponding mix proportion of 3% CCS and 4% FA (CCSFA4) was also theoretically chosen for the dry-wet cycle test.

Fig 11 shows the apparent states of stabilized soils through 5 cycles. After the first and second cycles, the surfaces of all specimens remained relatively intact without any obvious cracks. However, after the third and fourth cycles, varying degrees of spalling appeared on the surface of both stabilized soils. The specimens from the QLFA4 exhibited severe surface spalling and distinct cracks, whereas the CCSFA4 specimens showed only slight spalling at the bottom. After the final cycle, specimens from the QLFA4 were directly damaged. It was obvious that the CCS-FA stabilized soil demonstrated superior durability.

To explore the durability of stabilized soil, the variations in height and mass of specimens were recorded at the end of each cycle (Fig 12). The mass loss rates of both stabilized soils remained within 1% at the first cycle. However, there was a significant increase in mass loss rates for both stabilized soils after the third cycle. The rates reached 3.39% for QLFA4 and 1.50% for CCSFA4, with corresponding height changes of 0.38mm and 0.21mm, respectively. In contrast, the mass loss and height change of QLFA4 specimens were greater than those of CCSFA4. By the end of the fifth cycle, the QLFA4 specimens were directly damaged, whereas the height variation of the CCSFA4 specimens tended to stabilize. A trend could be found that in early cycles, the decrease is very severe and lighter in later cycles. As the cycles progressed, the specimens underwent a process of water loss and contracted followed by water absorption and expansion, resulting in the continuous deterioration of stabilized soil. The increase in spalling after immersion caused a reduction in height and mass loss. It was noteworthy that throughout the dry-wet cycle test, the mass loss rate and height change of the CCSFA4 specimens were consistently lower than those of the QLFA4 specimens.

The variation of the UCS of the specimens under dry-wet cycles is shown in Fig 13. The specimens under the standard curing conditions (Control Group) are represented by CG. It was obvious that the UCS decreases with the cycle number. The impact of dry-wet cycles on

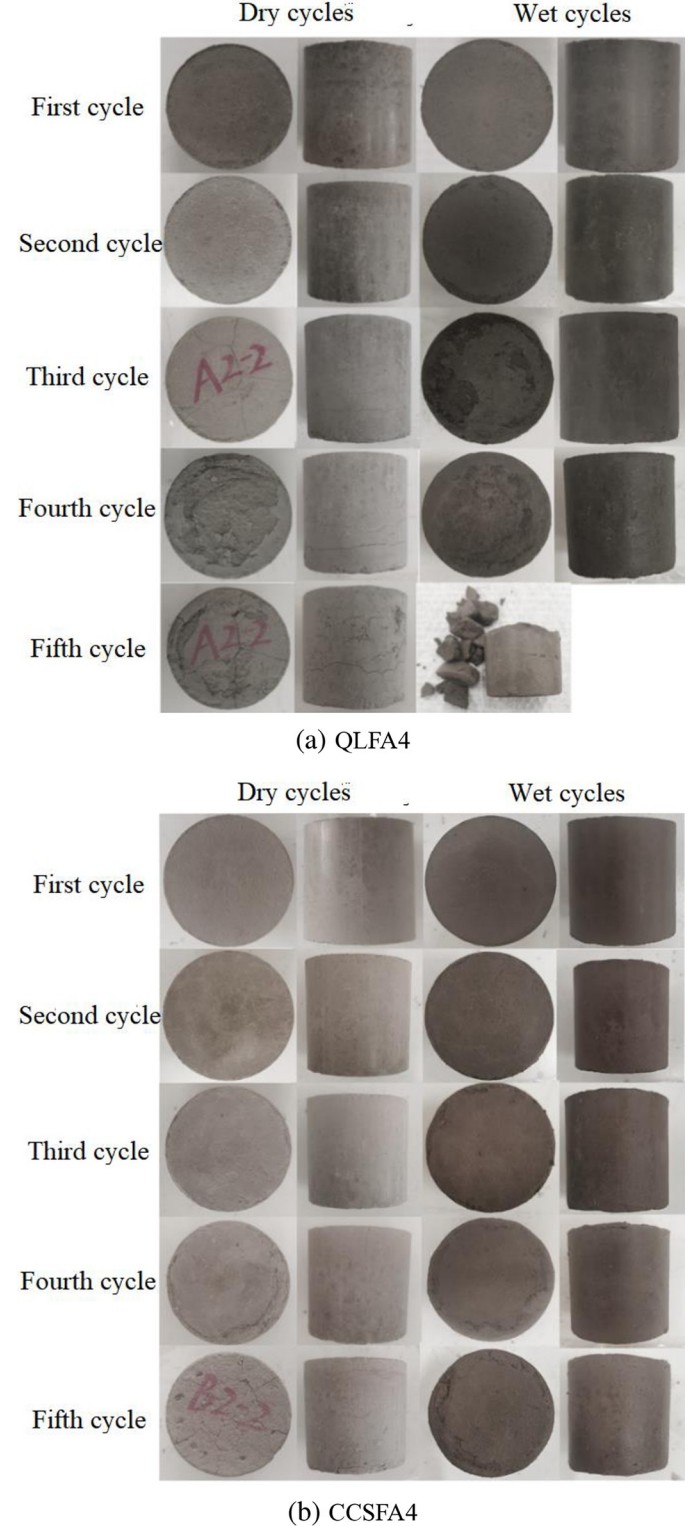

(a) QLFA4

(b) CCSFA4

**Fig 11. The apparent state of specimens under dry-wet cycles.**

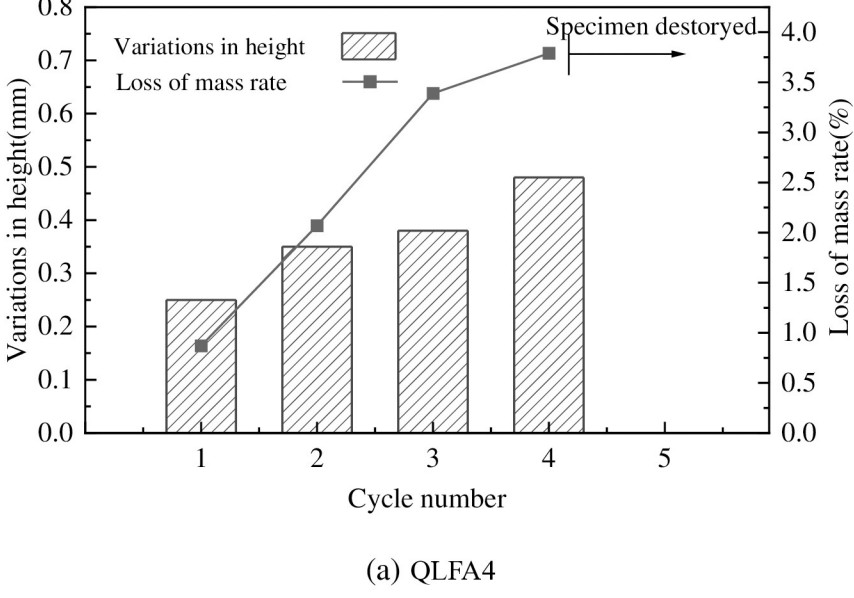

(a) QLFA4

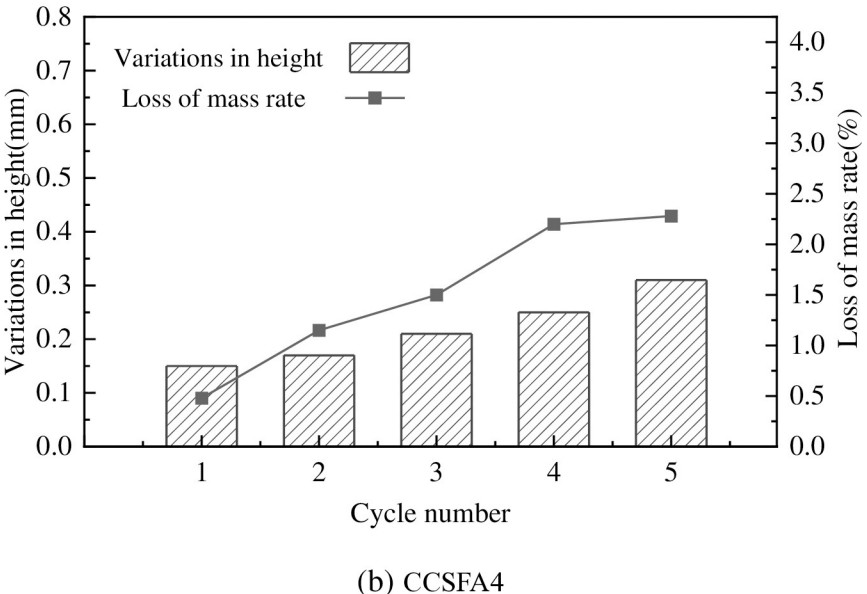

(b) CCSFA4

**Fig 12. Variation curve of mass loss rate and height changes of specimens under dry-wet cycles.**

QL-FA stabilized soil was greater than that on CCS-FA stabilized soil, with the strength decreasing by 36.50% and 54.90% at the first and third cycles, respectively. After five cycles, the specimens from QLFA4 were destroyed. For CCS-FA stabilized soil, the UCS reductions after the corresponding numbers of dry-wet cycles are only 26.30%, 36.96%, and 55.29% respectively.

The repeated immersion and evaporation of water within the specimens disrupted their original structure, resulting in a decrease in UCS. Meanwhile, the difference in moisture content between the interior and exterior of the specimens generated tensile stresses on the specimen's surface. When the cohesion between particles is insufficient to resist these tensile

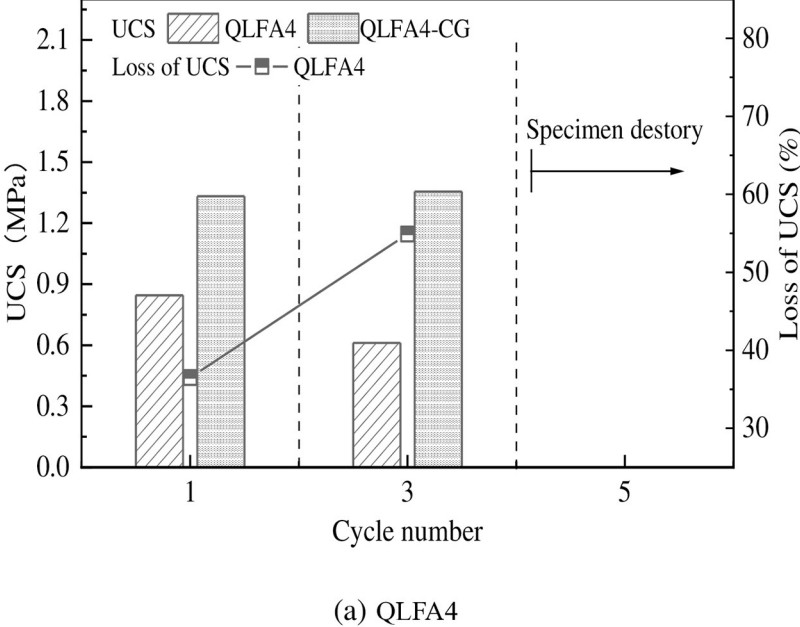

(a) QLFA4

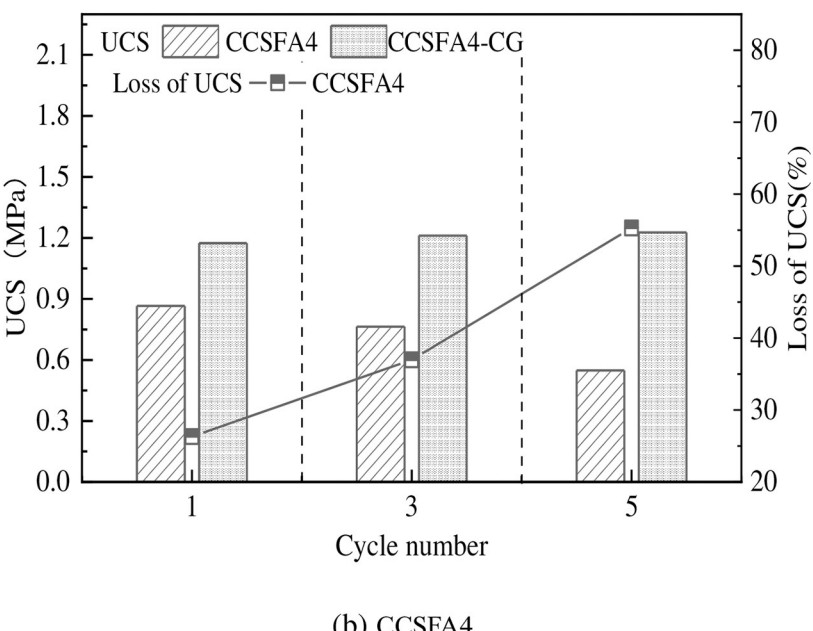

(b) CCSFA4

**Fig 13. Variation curve of UCS of specimens under dry-wet cycles.**

stresses, cracks form within the specimens [36]. As the cycle number increases, the expansion of microcracks and the redistribution of hydration products intensifies, leading to the destruction of soil structure [37]. In comparison with the CCS-FA stabilized soil, the UCS of QL-FA stabilized soil decreased significantly, which is consistent with the apparent state, mass loss, and height changes of the specimens. Hence, it can be concluded that the durability of CCS-FA stabilized soil is superior to QL-FA stabilized soil. Using of CCS as a substitute for QL

could enhance the durability of the stabilized soil. In general, the improvement is mainly due to the finer particle size and larger specific surface area of CCS. The CCS plays a good filling role and makes a denser microstructure of the stabilized soil, leading to greater erosion resistance and chemical stability [17,20], consequently enhancing the durability of the soil under dry-wet cycle conditions. Moreover, with the admixture of CCS, the charge state of the shield residue is altered. A more compacted soil particle cementitious system is formed, thereby further enhancing the durability of the stabilized soil.

## 4. Conclusion

To achieve the utilization of discarded shield residue and alleviate the shortage of subgrade filling, CCS, QL, and FA were selected to enhance the properties of shield residue. The mechanical properties of CCS-FA stabilized soil and QL-FA stabilized soil were compared. The feasibility of CCS to improve shield residue as subgrade filling was validated. The main conclusions are as follows:

1. The mechanical properties of the shield residue with additive contents (QL-FA stabilized soil and CCS-FA stabilized soil) exhibited better performance in CBR values. Initially, the CBR values of CCS-FA stabilized soil were slightly lower than those of QL-FA stabilized soil, but after 28 days, they exceeded them by more than 200%.

2. With the increase in curing time, the UCS and water stability improved significantly. However, the moisture content and pH value of the stabilized soil decreased. The early UCS of CCS-FA stabilized soil was slightly lower than that of QL-FA stabilized soil. After 60 curing days, all stabilized soil exhibited a UCS value exceeding 1.9 MPa.

3. Under the same curing time, the water stability coefficient ($k$) of the CCS-FA stabilized soil was higher than that of QL-FA stabilized soil. Especially at 7 days and 14 days, the $k$ of the CCS-FA stabilized soil was approximately 1.05 to 1.26 times that of the QL-FA stabilized soil.

4. With the increase in cycle numbers, the UCS of both CCS-FA stabilized soil and QL-FA stabilized soil decreased significantly. Spalling degrees on specimen surfaces gradually increased, accompanied by a substantial increase in the mass loss rate. After the fifth dry-wet cycle, the QL-FA stabilized soil was directly destroyed, while the CCS-FA stabilized soil maintained overall integrity.

5. Substituting CCS for QL initially resulted in a slight decrease in both the CBR values and UCS of the specimens. However, the long-term strength of the CCS-FA stabilized soil became comparable to that of the QL-FA stabilized soil. Importantly, using CCS instead of QL enhanced the water stability and durability of the stabilized soil. The CCS could effectively substitute QL to enhance the mechanical properties of shield residue as subgrade filling.

Future investigations will delve into the following aspects for exploration. First, the dry-wet cycle test will be carried out on specimens of all mix proportions to further explore the effect of QL substitution by CCS and increasing addition of FA. Then, performing microscopic tests such as scanning electron microscopy and thermal gravimetric analysis to gain a deeper understanding of the soil's microstructure and mechanical mechanism. In addition, the study will investigate the feasibility of using soda residue, flue gas desulfurization gypsum, or a combination of diverse supplementary cement materials in conjunction with CCS as substitutes for lime in subgrade filling.

## Supporting information

**S1 Data. The supplementary information (date) is provided in one single XLS file.**
(XLS)

## Author Contributions

**Conceptualization:** Jianwen Ding.

**Formal analysis:** Shoujie Wang, Qingying Guo, Chenhao Li.

**Funding acquisition:** Jianwen Ding.

**Investigation:** Shoujie Wang, Qingying Guo, Ning Jiao.

**Methodology:** Shoujie Wang, Ning Jiao.

**Supervision:** Jianwen Ding.

**Visualization:** Qingying Guo.

**Writing – original draft:** Shoujie Wang.

**Writing – review & editing:** Ning Jiao, Chenhao Li.

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
