## [Decision Letter · Decision Letter 0]

22 Oct 2024

PONE-D-24-34216

Mechanical properties of discarded shield residue improved by calcium carbide slag and fly ash as subgrade filling

PLOS ONE

Dear Dr. Ding,

Thank you for submitting your manuscript to PLOS ONE. After careful consideration, we feel that it has merit but does not fully meet PLOS ONE’s publication criteria as it currently stands. Therefore, we invite you to submit a revised version of the manuscript that addresses the points raised during the review process.

This manuscript has been reviewed by expert referees on this subject and it has been concluded that the draft article proposes an innovative idea for recycling Engineering waste and has certain research value, but it should be further improved, especially the effects of replacing QL with CCS or increasing FA on its properties. It has been concluded that it should be revised and resubmitted in line with the comments and suggestions of the referees below.

We look forward to receiving your revised manuscript.

Kind regards,

Ugur Ulusoy, Ph.D.

Academic Editor

PLOS ONE

Journal Requirements:

3. Thank you for stating the following financial disclosure: “This study is partially supported by the National Natural Science Foundation of China

 (Grant No. 52378330 and Grant No. 51978159).”

4. We note that your Data Availability Statement is currently as follows: “All relevant data are within the manuscript and in Supporting Information files.”

Please confirm at this time whether or not your submission contains all raw data required to replicate the results of your study. Authors must share the “minimal data set” for their submission. PLOS defines the minimal data set to consist of the data required to replicate all study findings reported in the article, as well as related metadata and methods (https://journals.plos.org/plosone/s/data-availability#loc-minimal-data-set-definition). For example, authors should submit the following data: - The values behind the means, standard deviations and other measures reported; - The values used to build graphs; - The points extracted from images for analysis. Authors do not need to submit their entire data set if only a portion of the data was used in the reported study. If your submission does not contain these data, please either upload them as Supporting Information files or deposit them to a stable, public repository and provide us with the relevant URLs, DOIs, or accession numbers. For a list of recommended repositories, please see https://journals.plos.org/plosone/s/recommended-repositories. If there are ethical or legal restrictions on sharing a de-identified data set, please explain them in detail (e.g., data contain potentially sensitive information, data are owned by a third-party organization, etc.) and who has imposed them (e.g., an ethics committee). Please also provide contact information for a data access committee, ethics committee, or other institutional body to which data requests may be sent. If data are owned by a third party, please indicate how others may request data access.

5. Please amend either the title on the online submission form (via Edit Submission) or the title in the manuscript so that they are identical.

Reviewers' comments:

Reviewer's Responses to Questions

**Comments to the Author**

1. Is the manuscript technically sound, and do the data support the conclusions?

Reviewer #1: Yes

Reviewer #2: Yes

2. Has the statistical analysis been performed appropriately and rigorously? 

Reviewer #1: Yes

Reviewer #2: Yes

3. Have the authors made all data underlying the findings in their manuscript fully available?

Reviewer #1: Yes

Reviewer #2: Yes

4. Is the manuscript presented in an intelligible fashion and written in standard English?

Reviewer #1: Yes

Reviewer #2: Yes

5. Review Comments to the Author

Reviewer #1: PONE-D-24-34216:

Comments and Suggestions

This manuscript aims to enhance the shield residue with IBPs (i.e. CCS and FA) and utilize it for subgrade filling during the construction of Jihua Road, in order to achieve resource utilization and cost savings. The study presents several techniques to investigate the properties of the samples; the results were also explained and discussed. However, the quality of the manuscript should be further enhanced, particularly the effects of QL substitution by CCS or increasing FA on their properties. The clarifying and potential points are as follows:

Major comments:

Affiliation

- The superscript of affliliation for all authors should be corrected.

Introduction

1) In litterature reviews with CCR and FA for stabilization soil, plese consider the related recently publications such as :

- Environmentally friendly binders from calcium carbide residue and silica fume and feasibility for soft clay stabilization, https://doi.org/10.1016/j.cscm.2024.e03117

- Evaluation of calcium carbide residue and fly ash as sustainable binders for environmentally friendly loess soil stabilization, https://www.nature.com/articles/s41598-024-51326-x,

- Enhancing durability of concrete mixtures with supplementary cementitious materials: A study on organic acid corrosion and physical abrasion in pig farm environments. https://doi.org/10.1016/j.cscm.2023.e02731

- Acidic corrosion-abrasion resistance of concrete containing fly ash and silica fume for use as concrete floors in pig farm, https://www.sciencedirect.com/science/article/pii/S2214509522001425?via%3Dihub

Materials and methods

2) In XRD patterns, the scale of intensity (Y axis) should be provided and also it should be disscus on the level of crystallity or amprphous phases for all raw materials.

3) There appears to be an inconsistency in using CCS in this study, as the specimen's abbreviation is CCR3FA.

4) Please provide the reason or evidence to support the appropriated experimental misproportions.

Results and discussion

5) Please make sure to thoroughly review and correct the data presented in Table 4. Additionally, kindly provide an explanation and analysis of the results depicted in the table, which indicate variations in moisture content due to the effects of QL substituted by CCS or increasing FA.

6) Please cafully check and correct the data in table 4. Also, please explian and discuss the results in this table which is observed that the moiture content is varied based on the effects of of QL substituted by CCS or increasing of FA.

7) In the CBR test, the results of all specimens should be presented, and the effect of QL substituted by CCS or increasing FA should be discussed.

8) In the UCS test, “After 28 days, the strength growth of the QL-FA stabilized soil slowed down, while that of the CCS-FA stabilized soil continued to progress at a fast pace.” Please explain the reasons for or provide results to support the mechanisms of CCS-FA stabilized soil.

9) Please provide and further discuss the significant results obtained from equation 1, table 5, and Fig. 6.

10) In the moisture content and pH test, please explain the reason to support that those values of CCR are larger than QL specimens.

11) In the water stability test, “This phenomenon could own to the finer particle size and larger specific surface area of CCS, which could lead to a denser microstructure of the stabilized soil.” Please provide the supported results, such as particle size and morphology of QL, CCS, and FA as investigated by SEM measurement.

12) In the Dry-wet cycle test, please explain why the QL3FA4 and CCR3FA4 specimens were selected for testing over other appropriate specimens.

13) In the Dry-wet cycle test, Please provide the results of all specimens in the Dry-wet cycle test and discuss the effects of QL substitution by CCS or increasing FA on this test while enhancing shield residue.

14) In the Dry-wet cycle test, “In comparison with the CCS-FA stabilized soil, the UCS of QL-FA stabilized soil decreased significantly, which is consistent with the apparent state, mass loss, and height changes of the specimens. Hence, it can be concluded that the durability of CCS-FA stabilized soil is superior to QL-FA stabilized soil”. Please provide the results of all speciements and carfully dissussion on the effects of QL substitution by CCS or increasing FA.

15) In the Dry-wet cycle test, “In general, the improvement is mainly due to the finer particle size and larger specific surface area of CCS. The CCS plays a good filling role and makes a denser microstructure of the stabilized soil, leading to greater erosion resistance and chemical stability” I The results of this study have not been supported; please provide the particle size and morphology of QL, CCS, and FA as investigated by SEM measurement.

Conclusions

1) Please summarize the impact of QL substitution by CCS or increasing FA on this study.

2) What are the future studies and proposed applications of shield residual based on their properties?

Reviewer #2: This manuscript comprehensively analyzes the improvement of mechanical properties of shield residue by calcium carbide slag and fly ash, and obtains some reliable results. An innovative idea of recycling engineering waste is proposed, which has certain research value, but there are still some deficiencies. The comments and suggestions are as follows.

1. Sec. 3.3, there is a lack of strength and other mechanical parameters of shield muck samples, it is recommended to be supplemented.

2. Sec.3.3, the reaction trends of QL-FA and CCR-FA are not the same, please explain further.

3. Sec.3.1, line 117, please explain why the higher the content of FA, the smaller the particle size, or give the specific steps for making samples with different proportions?

4. Only the results of 3 % CCS and 3 %, 4 %, 6 % FA content are analyzed, are they representative?

5. In section of Conclusion, It is suggested to explain the limitations of the research, as well as the prospect and improvement direction of the research.

6. PLOS authors have the option to publish the peer review history of their article (what does this mean?). If published, this will include your full peer review and any attached files.

Reviewer #1: No

Reviewer #2: No

---

## [Author Response · Author response to Decision Letter 0]

28 Oct 2024

Response to the comments of Editor and Reviewers

Manuscript ID: PONE-D-24-34216

Title: Mechanical properties of discarded shield residue improved by calcium carbide slag and fly ash

Authors: Shoujie Wang, Jianwen Ding, Qingying Guo, Ning Jiao, Chenhao Li

Thank you very much for your nice suggestions and comments on our manuscript. There is no doubt that these suggestions and comments are valuable and very helpful for revising and improving our manuscript.

We have revised our manuscript carefully and accordingly with regard to all the comments and the suggestions, and the amendments are highlighted in blue in the revised manuscript. The responses to the comments of reviewers are detailed below, in which the paragraphs in normal fonts are the comments and the authors’ responses are written in bold fonts. 

Response to Editor’s Comments

Answer:

Thank you for your guidance on ensuring that our manuscript adheres to the style requirements of PLOS ONE. We have carefully reviewed and updated our manuscript as requested. Thank you again for your feedback and for providing the resources to assist us in this process. 

2.Please note that funding information should not appear in any section or other areas of your manuscript. We will only publish funding information present in the Funding Statement section of the online submission form. Please remove any funding-related text from the manuscript.

Answer:

Thank you for your instruction regarding the inclusion of funding information in our manuscript. In response to your request, we have thoroughly reviewed our manuscript and have removed all funding-related text from the manuscript. 

3.Thank you for stating the following financial disclosure: “This study is partially supported by the National Natural Science Foundation of China (Grant No. 52378330 and Grant No. 51978159).” Please state what role the funders took in the study. If the funders had no role, please state: "The funders had no role in study design, data collection and analysis, decision to publish, or preparation of the manuscript." If this statement is not correct you must amend it as needed. Please include this amended Role of Funder statement in your cover letter; we will change the online submission form on your behalf.

Answer:

Thank you for your suggestion. In accordance with your instructions, we have included the following statement in our cover letter to ensure the funders' role:

Thank you again for your assistance with this matter.

4.We note that your Data Availability Statement is currently as follows: “All relevant data are within the manuscript and in Supporting Information files.” Please confirm at this time whether or not your submission contains all raw data required to replicate the results of your study. Authors must share the “minimal data set” for their submission.

Answer:

Thank you for pointing out. In response to your comment, we have compiled the relevant data into an Excel file, and thank you again for your assistance with this matter.

5.Please amend either the title on the online submission form (via Edit Submission) or the title in the manuscript so that they are identical.

Answer:

Thank you for your observation regarding the consistency of the title in our manuscript and the online submission form. Upon receiving your feedback, we have reviewed the title in the manuscript to match the title on the online submission form. We appreciate your attention to detail and thank you for your assistance.

Response to Reviewer #1’s Comments

This manuscript aims to enhance the shield residue with IBPs (i.e. CCS and FA) and utilize it for subgrade filling during the construction of Jihua Road, in order to achieve resource utilization and cost savings. The study presents several techniques to investigate the properties of the samples; the results were also explained and discussed. However, the quality of the manuscript should be further enhanced, particularly the effects of QL substitution by CCS or increasing FA on their properties. The clarifying and potential points are as follows:

Affiliation

1.The superscript of affliliation for all authors should be corrected.

Answer: 

Thank you for your attention to detail and for pointing out the need for correction in our manuscript. We apologize for the oversight regarding the superscript of affiliation for our authors. Upon receiving your feedback, we have carefully reviewed and corrected the superscript affiliations for all authors to ensure accuracy and consistency throughout the paper.

Introduction

2.In litterature reviews with CCS and FA for stabilization soil, please consider the related recently publications such as:

- Environmentally friendly binders from calcium carbide residue and silica fume and feasibility for soft clay stabilization, https://doi.org/10.1016/j.cscm.2024.e03117

- Evaluation of calcium carbide residue and fly ash as sustainable binders for environmentally friendly loess soil stabilization, 

https://www.nature.com/articles/s41598-024-51326-x,

- Enhancing durability of concrete mixtures with supplementary cementitious materials: A study on organic acid corrosion and physical abrasion in pig farm environments. https://doi.org/10.1016/j.cscm.2023.e02731

- Acidic corrosion-abrasion resistance of concrete containing fly ash and silica fume for use as concrete floors in pig farm, 

https://www.sciencedirect.com/science/article/pii/S2214509522001425?via%3Dihub

Answer: 

Thank you for your suggestion to include more recent high-quality publications in our paper. We have added these in the appropriate location. This suggestion makes the structure of the article more complete. These references are highly valuable, and we have discussed and referenced them in the introduction section. We believe that these revisions have significantly enhanced the depth and breadth of our paper (please refer to reference [13], [14], [18], and [22]).

Materials and methods

3.In XRD patterns, the scale of intensity (Y axis) should be provided and also it should be discus on the level of crystallity or amprphous phases for all raw materials.

Answer: 

Thank you for your insightful comments and suggestions. We appreciate your attention to the details of our X-ray diffraction (XRD) analysis. In response to your recommendation, we have now included the scale of intensity on the Y-axis in the XRD patterns presented in our manuscript. This addition will allow readers to more accurately interpret the data and compare the relative intensities of the peaks. Furthermore, we have added our discussion to include an analysis of crystallinity and amorphous phases for all raw materials. (please see lines 68-72)

The additions are as follows:

Based on the XRD patterns shown in Fig.1, the primary crystalline material in the shield residue was calcite (CaCO3) and quartz (SiO2). And the predominant crystalline material in both QL and CCS was portlandite (Ca(OH)2). As for FA, the broad hump suggested that FA is mainly comprised of amorphous phases with some peaks of Chiastolite (Al2(SiO2)O), quartz, ferric oxide, and lime (please refer to Lines 80-85).

4.There appears to be an inconsistency in using CCS in this study, as the specimen's abbreviation is CCS3FA.

Answer: 

Thank you for pointing this out. The dosage of CCS and QL in this study is consistent. Upon your suggestion, we have confirmed their consistency throughout the experiments. Sorry for causing trouble, we have now corrected the specimen abbreviations from CCS3FA to CCSFA and from QL3FA to QLFA to accurately reflect the content of the material in the paper.

5.Please provide the reason or evidence to support the appropriated experimental misproportions.

Answer: 

Thank you for your inquiry regarding the selection of experimental proportions in our study. In our research, we have carefully considered the selection of our mix proportions before the experiment. The proportions of FA used in our experiments were determined based on preliminary tests and the requirements for subgrade materials. During these pre-experiments, we found that a 2% FA content did not meet the necessary strength and durability for subgrade materials. It can be seen from Fig. 2 and Fig. 4, that the CBR values and UCS of all stabilized soil surpassed the requirement specifications for the design of highway subgrades. Additionally, in line with economic considerations, we have chosen 3% lime with 3%, 4%, and 6% for the experiment. The corresponding mix proportion of 3% CCS with varying percentages of FA (3%, 4%, and 6%) was also theoretically chosen for the experiment.

Results and discussion

6.Please make sure to thoroughly review and correct the data presented in Table 4. Additionally, kindly provide an explanation and analysis of the results depicted in the table, which indicate variations in moisture content due to the effects of QL substituted by CCS or increasing FA.

Answer: 

Thanks to the Reviewer’s comment. We appreciate your request for a thorough examination of the data and an in-depth analysis of the results. In response to your feedback, we have carefully reviewed and made the necessary corrections to the data in Table 4 to ensure its accuracy. We have also added and corrected the analysis of the variations of optimum moisture content and maximum dry density of stabilized soil in Section 3.1 (please refer to Lines 107-119).

The revised content is as follows:

It could be observed that after adding improved materials to the shield residue, the optimum moisture content of the stabilized soil increased, while the maximum dry density decreased somewhat. This might be attributed to the lower density of the improved material compared to shield residues, as well as some flocculent cementation products that result in a certain volume expansion, thereby reducing soil compactness and density [34-35]. Conversely, the incorporation of FA could fill soil pores and the greater specific surface area of FA could facilitate the formation of bound water as water molecules infiltrate the micropores [15]. As a result, the optimal water content of stabilized soil increased with the increase of FA [20, 36]. Besides, FA was prone to pozzolanic reactions with QL and CCS to generate cementitious materials. This reaction consumed a large amount of water, implying additional water needs to be added in order to obtain the stabilized soil with optimal water content. Consequently, as the FA content increased, the optimum moisture content of the stabilized soil also increased [16]

7.Please carefully check and correct the data in table 4. Also, please explian and discuss the results in this table which is observed that the moiture content is varied based on the effects of of QL substituted by CCS or increasing of FA.

Answer: 

Thank you for your reminder. We appreciate your request for a thorough examination of the data and an in-depth analysis of the results. In response to your feedback, we have carefully checked and corrected the date in Table4 to ensure its accuracy. And we have added the analysis of the variations of optimum moisture content and maximum dry density of stabilized soil in Section 3.1 (please refer to Lines 107-119). The revised content can refer to the response provided in question 7.

8.In the CBR test, the results of all specimens should be presented, and the effect of QL substituted by CCS or increasing FA should be discussed.

Answer: 

Thank you for your valuable suggestion regarding the presentation of CBR test results. We have taken your guidance seriously and have now included the CBR value and swelling rate of the raw shield residue, which were 8.1% and 3.0%, respectively, in Section 3.2 of our manuscript (please refer to lines 126-127). 

Furthermore, in line with your recommendation, we have checked and corrected the Fig. 2 and Fig. 3 to ensure that all CBR test results for the stabilized soil are presented. Additionally, we have expanded and corrected Section 3.2 to include a more detailed discussion on the effects when QL is substituted by CCS and with the increasing addition of FA . 

The effects on QL are substituted by CCS and with the increasing addition of FA can be concluded as follows (please refer to lines 135-142 and lines 148-152):

These CBR values of CCS-FA stabilized soil were slightly lower than those of the QL-FA stabilized soil at the same FA content. The incorporation of FA could also enhance the bearing capacity of CCS-FA stabilized soil and QL-FA stabilized soil. Furthermore, the swelling rate of the QL-FA stabilized was slightly lower compared to CCS-FA stabilized soil, but the difference between the two is small, which indicated that the swelling trend of both stabilized soils was not significantly different.

The impact of curing time on the CBR values was slightly lower at the initial stage for the CCS-FA stabilized soil than that of the QL-FA stabilized soil. However, the CBR values exceeded those of the QL-FA stabilized soil slightly at 28 days. In addition, the swelling rate of both stabilized soils substantially decreased, reducing to within 0.1% after 7 days of standard curing.

9.In the UCS test, “After 28 days, the strength growth of the QL-FA stabilized soil slowed down, while that of the CCS-FA stabilized soil continued to progress at a fast pace.” Please explain the reasons for or provide results to support the mechanisms of CCS-FA stabilized soil.

Answer: 

Thank you for pointing this out. We have analyzed the strength curves presented in Figure 4, which reveal that while the QL-FA stabilized soil exhibited a rapid strength increase in the first 28 days, its rate of strength gain slowed down subsequently. In contrast, the CCS-FA stabilized soil maintained a more sustained pace of strength development beyond the initial 28 days. 

To provide a clearer understanding, we have calculated the average daily strength growth rates for both types of stabilized soils. For the QL-FA stabilized soil, the rate was 16.0 kPa per day during the first 28 days and slowed to 1.9 kPa per day from days 28 to 60. On the other hand, the CCS-FA stabilized soil showed a growth rate of 11.4 kPa per day in the first 28 days, which decreased but remained higher at 6.0 kPa per day from days 28 to 60. This phenomenon indicates that the strength growth rate of CCS-FA stabilized soil is greater than that of QL-FA stabilized solidified soil after initial 28 days.

Sorry for causing trouble, we have addad the reasons of the phenomenon in Section 3.3 (please refer to Lines 181-184).

10.Please provide and further discuss the significant results obtained from equation 1, table 5, and Fig. 6.

Answer: 

Thank you for your suggestion to further discuss the significant results obtained from Equation 1, Table 5, and Figure 6. In response to your feedback, we have incorporated a detailed discussion of the results derived from Equation 1, the data presented in Table 5, and the observations from Figure 6 within Section 3.3 of our manuscript (please refer to Lines 214-219).

The revised content is as follows:

The hyperbolic function could effectively fit the characteristic curves of UCS as it varies with the curing time. Analysis of the regression parameters revealed that the variation range for parameter p1 is relatively small and was not significantly affected by the type or content of the improved materials, remaining between 1.94 and 2.14. In contrast, variation in material content had a more significant impact on parameter p2, which ranges from 2.73 to 3.50.

11.In the moisture content and pH test, please explain the reason to support that those values of CCS are larger than QL specimens.

Answer: 

Thank you for your inquiry regarding the moisture content and pH values in the CCS-FA stabilized soil compared to the QL-FA stabilized soil. The var

---

## [Decision Letter · Decision Letter 1]

13 Nov 2024

Mechanical properties of discarded shield residue improved by calcium carbide slag and fly ash as subgrade filling

PONE-D-24-34216R1

Dear Dr. Ding,

We’re pleased to inform you that your manuscript has been judged scientifically suitable for publication and will be formally accepted for publication once it meets all outstanding technical requirements.

Kind regards,

Ugur Ulusoy, Ph.D.

Academic Editor

PLOS ONE

Additional Editor Comments (optional):

Following the completion of our expert reviewers' evaluation, it was decided to manuscript the article as it is, since it was seen that the Authors addressed the problems that emerged in the first review and made corrections in the study.

Reviewers' comments:

Reviewer's Responses to Questions

**Comments to the Author**

1. If the authors have adequately addressed your comments raised in a previous round of review and you feel that this manuscript is now acceptable for publication, you may indicate that here to bypass the “Comments to the Author” section, enter your conflict of interest statement in the “Confidential to Editor” section, and submit your "Accept" recommendation.

Reviewer #1: All comments have been addressed

Reviewer #2: (No Response)

2. Is the manuscript technically sound, and do the data support the conclusions?

Reviewer #1: Yes

Reviewer #2: Yes

3. Has the statistical analysis been performed appropriately and rigorously? 

Reviewer #1: Yes

Reviewer #2: Yes

4. Have the authors made all data underlying the findings in their manuscript fully available?

Reviewer #1: Yes

Reviewer #2: Yes

5. Is the manuscript presented in an intelligible fashion and written in standard English?

Reviewer #1: Yes

Reviewer #2: Yes

6. Review Comments to the Author

Reviewer #1: The authors have considered all recommended requests & addressed the various precisions both scientifically & technically.

Reviewer #2: The paper has been improved with revisions and the authors have addressed the concerns raised in the initial review.

7. PLOS authors have the option to publish the peer review history of their article (what does this mean?). If published, this will include your full peer review and any attached files.

Reviewer #1: No

Reviewer #2: No

---

## [Editor Report · Acceptance letter]

20 Nov 2024

PONE-D-24-34216R1 

PLOS ONE

Dear Dr. Ding, 

I'm pleased to inform you that your manuscript has been deemed suitable for publication in PLOS ONE. Congratulations! Your manuscript is now being handed over to our production team.

Kind regards, 

on behalf of

Prof. Dr. Ugur Ulusoy 

Academic Editor

PLOS ONE